

# Stinging wasps (Hymenoptera: Aculeata), which species have the longest sting?

Emily A. Sadler[1], James P. Pitts[1] and Joseph S. Wilson[2]

[1] Department of Biology, Utah State University, Logan, UT, USA
[2] Department of Biology, Utah State University—Tooele, Tooele, UT, USA

## ABSTRACT

The stings of bees, wasps, and ants are something that catches the attention of anyone that experiences them. While many recent studies have focused on the pain inflicted by the stings of various stinging wasps, bees, or ants (Hymenoptera: Aculeata), little is known about how the length of the sting itself varies between species. Here, we investigate the sting length of a variety of aculeate wasps, and compare that to reported pain and toxicity values. We find that velvet ants (Hymenoptera: Mutillidae) have the longest sting compared to their body size out of any bee, wasp, or ant species. We also find that there is no link between relative sting length and pain; however, we did find an inverse relationship between relative sting length and toxicity with taxa having shorter relative stings being more toxic. While we found a significant relationship between host use and relative sting length, we suggest that the long sting length of the velvet ants is also related to their suite of defenses to avoid predation.

## INTRODUCTION

Hymenoptera (ants, bees, and wasps) have likely been of interest to humans for as long as we have existed. Our histories are tied closely together with Hymenoptera. Paleolithic paintings depicting bees have been found dating from 15,000 years ago in the caves of Spain, and pottery vessels used for beekeeping have been found dating from 9,000 years ago (*Piek, 1986*; *Roffet-Salque et al., 2015*). However, as we have sought out honey and wax for our own benefit during the Meso- and Neolithic, we also were undoubtedly introduced to bee's defensive stings. While colloquially many refer to the "stinger" of bees, aculeate wasps, and ants, among entomologists the correct term both for the stinging structure (the noun) and the action (the verb) is simply "sting" and as entomologists we will refer to the structure as such. For example, it would be correct to say "the bee was able to sting me with her sting" rather than "... sting me with her stinger." Bees, ants, and many of the more familiar wasps fall within an infraorder of Hymenoptera called Aculeata, which is defined by the modification of the egg laying device (ovipositor) into a sting apparatus. Among the aculeates, the use of the sting is primarily for prey capture or host paralysis, and defense. While people often associated stings with bees, many wasps, and ants have even more painful stings than bees (*Schmidt, 2016*).

Corresponding author
Joseph S. Wilson,
joeswilson@gmail.com

Hymenoptera venom is the most potent of any of the animal venom (*Schmidt, 1990a*). An estimated 100 deaths per year can be attributed to stinging Hymenoptera, which is three to four times the number of deaths that occur by snake bites (*Schmidt, 1986a*). However, with the exception of allergic reactions, most people only experience temporary pain and edema. In fact, an adult human could safely withstand 1,000 bee stings (*Fitzgerald & Flood, 2006*). It is estimated that a lethal dose does not occur until a threshold of 20 stings/kg (*Fitzgerald & Flood, 2006*). Although the result of a large number of stings may not be death, pain is a certainty.

As many of us have experienced, the amount of pain that bee and wasp stings cause varies by species. Yet for some reason, a bee sting (generally the sting of a European honey bee, *Apis mellifera*) has become somewhat of a benchmark for pain. A shot at the doctor's office, for example, is often equated to a "quick sting, no worse than a bee." *Starr (1985)*, with the later expansion by *Schmidt (1990a, 2016)*, created a pain scale that opens the topic of sting pain up to a more general audience. This method ranks pain on a *scale* that varies from one, which is the least painful, and includes small sweat bees (*Lasioglossum* spp.) and native fire ants (*Solenopsis geminata*), to four, which is the most painful, and includes the bullet ant (*Paraponera clavata*) and tarantula hawks (*Pepsis* spp.) (*Schmidt, 2016*).

Pain from wasp envenomation can come from two sources with the first being from the chemical composition of venom itself. Hymenoptera venom can be two forms, alkaloid and proteinaceous (*Blum, 1981*; *Schmidt, Blum & Overal, 1986*). Hymenoptera venoms are known to vary in toxicity as given by studies of $LD_{50}$ (a measure of lethality) and enzymatic activity. *Schmidt, Blum & Overal (1980)* found that the $LD_{50}$ of various aculeate wasps varied from 0.25 mg/kg to 71 mg/kg with harvester ants (*Pogonomyrmex* spp.) having the most toxic venoms and velvet ants (*Dasymutilla klugii*) and the German hornet (*Paravespula germanica*) having the least toxic of the venoms. *Schmidt, Blum & Overal (1986)* found that aculeate venom also varies in eight different enzymes with species of velvet ants (*Dasymutilla lepeletierii*) once again having some of the lowest of the enzymatic activities.

The second cause of pain is due to mechanical damage from the sting puncturing tissue. Spider wasps (Pompilidae) and velvet ants seem to be paradoxical having a high pain rating, but a low $LD_{50}$ and weak enzymatic activity (*Schmidt, 1986a, 2004*; *Schmidt, Blum & Overal, 1980*). Observations have been made that the wasps having the highest pain indices on the Schmidt and Starr Pain Scale (*Schmidt, 1990a, 2016*; *Starr, 1985*) are often those with the largest bodies. The reason that these wasps cause intense pain may be due to the morphology of the sting itself. Much attention has been paid to the chemical components of venom, but little has been done concerning the morphology of the sting, and specifically its relative length. While comparisons of ovipositor length have been done for parasitoid wasps (*Townes, 1975*), no comparisons have been made among the stinging wasps, the aculeates.

In this study we investigate the relative length of the stings of various aculeate wasps and compare these lengths to known measures of toxicity, enzyme activity, and pain. A particular focus is given to the velvet ants (Mutillidae), because they are known to not only have a painful (yet relatively harmless) sting, but also, they are known to have an exceptionally long sting (*Schmidt, 2016*) (Fig. 1A).
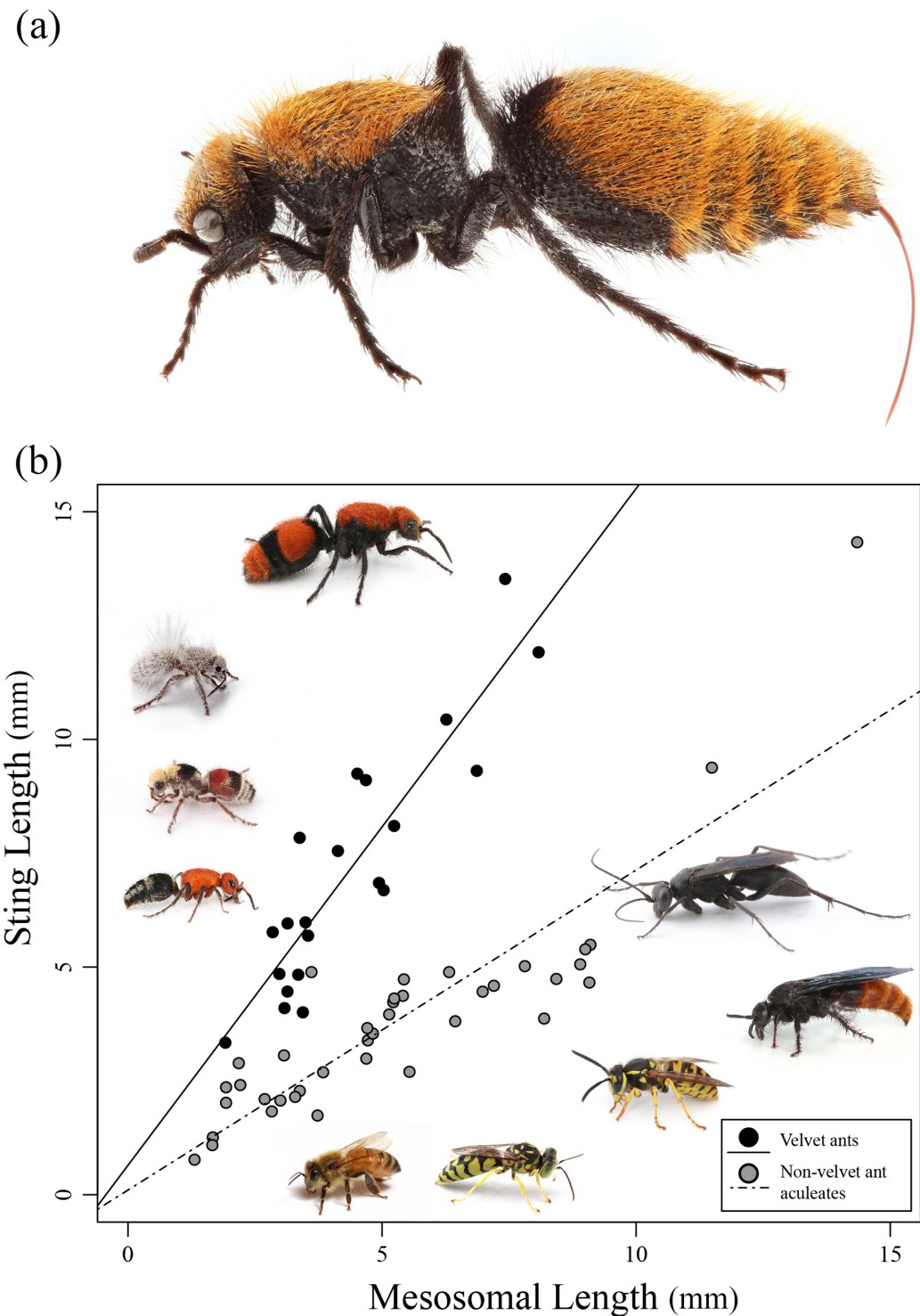

**Figure 1 Sting length vs. mesosomal length among aculeates.** (A) *Dasymutilla calorata* with her sting extended showing its length. (B) Graph of mesosomal length vs. sting length. Velvet ants are indicated in black and other aculeates in gray. Regression lines are indicated. Examples of various velvet ants and other aculeates are also pictured. Photo credit: Joseph S. Wilson.

## METHODS

### Sting length

The stings were measured for species from 14 families of aculeates (including ants and bees) (Table 1). All specimens were sourced from the Department of Biology Insect Collection at Utah State University (EMUS). The sting from each specimen was dissected and photographed using a Leica camera and microscope with light dome; calibrations were checked prior to any photographs. Measurements were taken from the tip of the lancet along the curve of the sting shaft to the beginning of the triangular plate using Image J (*Rasband, Image & U.S. National Institute of Health, 2011*). In order to obtain a relative measure of sting length to body size, the mesosomal length was measured as a proxy to overall body length. Because position of the head varies from specimen to specimen, and the gaster can be expanded or contracted depending on the specimen, total body lengths are difficult to determine and are not consistent from one individual to another. While various proxies have been used to estimate body length of Hymenoptera, including head width (*Haggard & Gamboa, 1980*), intertegular distance (*Greenleaf et al., 2007*), and wing length (*Bosch & Vicens, 2002*), proxies associated with wings could not be used, as not all aculeates have wings (i.e., ants and velvet ants). We selected the mesosoma as a proxy for body length, as it is not moveable and can easily be measured in preserved specimens. For a consistent measurement the mesosoma was measured from the anterior apex of the pronotal flange to the dorsal margin of the propodeal foramen in lateral view. Multiple images of each specimen were taken using a Leica camera with light dome. All images were then combined using Zerene Stacker v1.04, and measurements were made (based on 1 mm scale bar) using Image J (*Rasband, Image & U.S. National Institute of Health, 2011*).

Measurements were made in replicate depending on if the species was common. For common species, stings were extracted from five individuals and measurements of both the sting and the mesosoma were made for each specimen. These are indicated in Table 1 by those individuals with a standard deviation (STD) for both sting measurements and mesosomal length measurements. Because sting dissection is a destructive process, only a single specimen was extracted in instances where a species was rare. These species are indicated on Table 1 by those individuals without STD for either sting measurements or mesosomal length measurements.

We also wanted to make use of the extensive slide collection of aculeate stings at the EMUS. However, most stings previously slide-mounted had no associated voucher specimen, likely due to the destructive nature of sting extraction. To make use of these slides and avoid the destruction of additional museum specimens, for those species where a slide-mounted sting was available we selected five individuals of that same species from the EMUS collection and the mesosomal lengths were measured for these specimens. The largest and smallest specimens were measured, and a range of sizes between were chosen to represent a continuum. These are indicated in Table 1 by those individuals with STD for only the mesosomal length measurements.

**Table 1 Aculeate taxa included in analyses.**

| Family | Species | Sting | STD | Mesosoma | STD | Relative sting length | Sociality | Prey/host | Toxicity (LD50 mg/kg) | Pain |
|---|---|---|---|---|---|---|---|---|---|---|
| **Rhopalosomatidae** | Rhopalosomatidae nearcticum | 2.15 | | 3.29 | 0.23 | 0.65 | solitary | Grylloptera | | 1 |
| **Vespidae** | Vespa crabro | 5.39 | 0.28 | 9 | 0.57 | 0.6 | social | Predator | 2.9 | 2 |
| | Polistes apachus | 3.81 | 0.52 | 6.44 | 0.45 | 0.59 | social | Predator | 3.7 | 2.5 |
| **Tiphiidae** | Tiphia sp. 1 | 3.66 | | 4.71 | 0.78 | 0.78 | solitary | Coleoptera | | 1 |
| | Tiphia sp. 2 | 2.7 | 0.45 | 5.54 | 0.69 | 0.49 | solitary | Coleoptera | | 1 |
| **Chyphotidae** | Typhoctes peculiaris | 3.06 | | 3.07 | 1.01 | 1 | solitary | Unknown | | 1 |
| | Chyphotes mandibularis | 2.28 | | 3.38 | | 0.67 | solitary | Unknown | | 1 |
| | Chyphotes albipes | 1.26 | 0.24 | 1.67 | 0.31 | 0.75 | solitary | Unknown | | 1 |
| | Chyphotes belfragei | 1.83 | 0.3 | 2.83 | 0.48 | 0.65 | solitary | Unknown | | 1 |
| **Thynnidae** | Methoca stugia | 2.89 | | 2.18 | 0.50 | 1.32 | solitary | Coleoptera | | 1 |
| | Diamma bicolor | 3.87 | | 8.19 | 1.09 | 0.47 | solitary | Coleoptera | | 3 |
| | Anthobosca sp. | 2.1 | | 2.69 | 0.68 | 0.78 | solitary | Coleoptera | | 1 |
| | Myzinum sp. | 4.73 | | 5.43 | 0.61 | 0.87 | solitary | Coleoptera | | 1 |
| **Pompilidae** | Anoplius (Notiochares) lepidus | 5.49 | | 9.1 | 0.79 | 0.6 | solitary | Spiders | | 2.5 |
| | Anoplius (Pompilinus) insolens | 3.39 | | 4.72 | 0.51 | 0.72 | solitary | Spiders | | 2.5 |
| | Aporus luxus | 4.37 | | 5.41 | 0.19 | 0.81 | solitary | Spiders | | 2.5 |
| | Ageniella blaisdellii | 2.69 | | 3.84 | 0.44 | 0.7 | solitary | Spiders | | 2.5 |
| | Entypus unifasciatus | 4.74 | | 8.43 | 1.20 | 0.56 | solitary | Spiders | | 2.5 |
| | Evagetes sp. | 2.07 | | 3 | 0.57 | 0.69 | solitary | Spiders | | 2.5 |
| | Pepsis sp. | 14.33 | | 14.35 | 1.93 | 1 | solitary | Spiders | 65 | 4 |
| | Tachypompilus sp. | 4.66 | | 9.08 | 0.91 | 0.51 | solitary | Spiders | | 2.5 |
| | Sericopompilus sp. | 4.31 | | 5.24 | 0.56 | 0.82 | solitary | Spiders | | 2.5 |
| | Arachnospila arctus | 3.54 | | 4.82 | 0.54 | 0.73 | solitary | Spiders | | 2.5 |
| | Anoplius (Arachnophroctonus) chiapanas | 3.96 | | 5.14 | | 0.77 | solitary | Spiders | | 2.5 |
| **Sapygidae** | Sapyga elegans | 4.89 | 1.64 | 3.61 | 0.39 | 1.35 | solitary | Immature Hymenoptera | | 1 |
| **Myrmosidae** | Myrmosa unicolor | 2.41 | | 2.21 | 0.42 | 1.09 | solitary | Coleoptera | | 1 |
| **Mutillidae** | Atillum jucundum | 8.1 | | 5.24 | | 1.55 | solitary | Immature Hymenoptera | | 2 |
| | Cephalomutilla haematodes | 4.85 | | 2.98 | | 1.63 | solitary | Immature Hymenoptera | | 2 |
| | Dasylabris sp. | 7.84 | | 3.38 | | 2.32 | solitary | Immature Hymenoptera | | 2 |
| | Dasymutilla gloriosa | 9.11 | | 4.68 | | 1.94 | solitary | Immature Hymenoptera | | 2 |

(Continued)

| Family | Species | Sting | STD | Mesosoma | STD | Relative sting length | Sociality | Prey/host | Toxicity (LD50 mg/kg) | Pain |
|---|---|---|---|---|---|---|---|---|---|---|
| | *Dasymutilla nigripes* | 5.96 | | 3.14 | 0.90 | 1.9 | solitary | Immature Hymenoptera | | 2 |
| | *Dasymutilla occidentalis* | 13.52 | 0.98 | 7.42 | 0.54 | 1.82 | solitary | Immature Hymenoptera | 71 | 3 |
| | *Ephuta bellus* | 3.34 | | 1.92 | | 1.74 | solitary | Immature Hymenoptera | | 1 |
| | *Hoplomutilla phoreys* | 9.31 | | 6.86 | | 1.36 | solitary | Immature Hymenoptera | | 2 |
| | *Hoplomutilla xanthocerata* | 11.91 | | 8.08 | | 1.47 | solitary | Immature Hymenoptera | | 2 |
| | *Mutillina* sp. | 6.69 | | 5.03 | | 1.33 | solitary | Immature Hymenoptera | | 1.5 |
| | *Myrmilla erythrocephala* | 4.01 | | 3.44 | | 1.16 | solitary | Immature Hymenoptera | | 1.5 |
| | *Pertyella haracioi* | 4.83 | | 3.35 | | 1.44 | solitary | Immature Hymenoptera | | 1.5 |
| | *Pristomutilla* sp. | 4.1 | | 3.08 | | 1.33 | solitary | Immature Hymenoptera | | 1.5 |
| | *Pseudomethoca sanbornii* | 5.98 | | 3.49 | 0.77 | 1.71 | solitary | Immature Hymenoptera | | 1.5 |
| | *Pseudophotopsis komarovii* | 6.85 | | 4.94 | | 1.39 | solitary | Immature Hymenoptera | | 1.5 |
| | *Sigilla dorsata* | 4.46 | | 3.14 | | 1.42 | solitary | Immature Hymenoptera | | 1.5 |
| | *Smicromyrme viduata* | 9.25 | | 4.51 | | 2.05 | solitary | Immature Hymenoptera | | 1.5 |
| | *Sphaeropthalma pensylvanica* | 7.55 | | 4.13 | | 1.83 | solitary | Immature Hymenoptera | | 2 |
| | *Stenomutilla argentata* | 5.77 | | 2.85 | | 2.02 | solitary | Immature Hymenoptera | | 2 |
| | *Timulla grotei* | 5.69 | 0.3 | 3.54 | 0.21 | 1.61 | solitary | Immature Hymenoptera | | 2 |
| | *Tramautomutilla* sp. Paraguay | 10.44 | | 6.27 | | 1.67 | solitary | Immature Hymenoptera | | 3 |
| **Bradynobaenidae** | *Apterogyna* sp. | 2.36 | | 1.93 | 0.43 | 1.22 | solitary | Unknown | | 1 |
| | *Bradynobaenus* sp. | 2.02 | | 1.93 | | 1.05 | solitary | Unknown | | 1 |
| **Scolidae** | *Scoliia dubia dubia* | 5.06 | | 8.9 | | 0.57 | solitary | Coleoptera | | 1 |
| | *Camposomerus tolteca* | 4.59 | 0.4 | 7.2 | 0.6 | 0.64 | solitary | Coleoptera | 63 | 1 |
| **Formicidae** | *Paraponera clavata* | 4.89 | 0.14 | 6.32 | 0.6 | 0.77 | social | Predator | 6 | 4 |
| | *Solenopsis invicta* | 0.77 | 0.12 | 1.31 | 0.33 | 0.59 | social | Predator | | 1 |
| **Crabronidae** | *Bicyrtes caprioptera* | 4.23 | | 5.22 | 1.20 | 0.81 | solitary | Hemiptera | | 1 |
| | *Sphecius speciosus* | 9.38 | | 11.49 | 2.08 | 0.82 | solitary | Hemiptera | 46 | 1.5 |
| | *Oxybelus argentiopilosus* | 1.09 | | 1.66 | 0.44 | 0.66 | solitary | Diptera | | 1 |
| | *Astata bakeri* | 1.74 | | 3.73 | 0.13 | 0.47 | solitary | Hemiptera | | 1 |
| | *Bembix amoena* | 4.46 | 0.24 | 6.98 | 0.39 | 0.64 | solitary | Diptera | | 1 |
| **Apidae** | *Apis mellifera* | 2.99 | 0.13 | 4.69 | 0.39 | 0.64 | social | Herbivore | 2.8 | 2 |
| | *Xylocopa virginica* | 5.02 | 0.44 | 7.81 | 0.64 | 0.64 | solitary | Herbivore | 22 | 1 |

**Note:**
Sting and mesosomal lengths are given in mm and relative sting length is a ratio of sting length/mesosomal length. Sociality, host data, toxicity, and pain are derived from the literature (*Brothers & Finnamore, 1993*; *Schmidt, 1986a, 1986b, 1990a, 2004, 2016*; *Schmidt, Blum & Overal, 1980, 1986*; *Starr, 1985*). Toxicity data were not available for all taxa, but where possible, toxicity data for closely related species (within the same genus) was averaged for the genus and included in the table (e.g., *Polistes*). Similarly, pain data were not available for all taxa, but estimated pain values were included based on reported pain values, personal observation, or known pain indices from related taxa.

In addition to the sting measurements and mesosomal length measurements, we also calculated a relative sting length based on a ratio of the sting length to the mesosomal length.

## Toxicity, pain, and host preference

Measures of host preference, toxicity, and pain were derived from the literature (*Brothers & Finnamore, 1993*; *Schmidt, 1986a*, *1986b*, *1990a*, *2004*, *2016*; *Schmidt, Blum & Overal, 1980*, *1986*; *Starr, 1985*). Toxicity measures were only available for a handful of species (Table 1). For some species (e.g., *Polistes apachus*), toxicity measures were only available for closely related species within the same genus. In these cases we averaged the published toxicity of other members of the genus to estimate an average toxicity for these taxa. This averaging was only done at the genus level, so taxa without toxicity measures for other members of their genus were not included in the toxicity vs. sting length analyses.

While measuring pain from insect stings is undoubtedly a subjective endeavor, recently the "Schmidt pain index" has received much attention as it attempts to compare the pain of various Hymenoptera stings using a scale of one to four, 1 being low pain and 4 being high (*Schmidt, 2016*). Unfortunately, in addition to the pain rankings being subjective, they are generally not assigned to specific species, but rather given as a range for a taxonomic group (e.g., velvet ants, small species get a pain ranking of one to two, but no species identifications are given for these "small species" (*Schmidt, 1990a*)). To account for the lack of species-level measures of pain, we assigned each species we had a measure of sting length an estimated pain value based on the actual value for related species (when known). In many wasp families no measures of pain have been published, so for these instances we assigned potential pain values based on closely related wasp families and personal experience. Linear regression was used to compare the relative sting length to pain estimates.

## Data analysis

For the situations where there were multiple measurements for a given species, these measurements were then averaged and STDs were calculated (Table 1).

To investigate what factors were associated with the sting length, we used an ANCOVA with sting length as the response variable and mesosomal length, wasp type (velvet ant or other wasp), and the interaction between mesosomal length and wasp type as the predictor variables.

Furthermore, we used linear regression to compare sting length to various other measures. Toxicity measures (though only available for a subset of species) and pain estimates were individually compared to the relative sting length (sting length/mesosomal length) with relative sting length as the response variable, and either the pain estimates or the toxicity as the predictor variables. Sociality was also compared to relative sting length with relative sting length as the response variable and sociality (either social or solitary) predictor variables. Similarly sociality was compared to pain with pain being the response variable and sociality as the predictor variable. To determine if any trends existed in host choice, host data was compared to relative sting length with relative sting length
as the response variable and host as the predictor variable. All analyses were computed using R (*R Development Core Team, 2008*).

## RESULTS

### Sting length

Of the 21 species of velvet ants measured, the actual sting length ranged between 3.3 and 13.5 mm with *Dasymutilla occidentalis* having the longest sting. The relative sting length varied, but was above 1 for all velvet ants with a species of *Dasylabris* from Russia having the longest relative length. Of the 39 non-velvet ant wasps measured, the actual sting length varied between 1 and 14.3 mm with the relative sting length being below 1 for all but a few species (Table 1). *Pepsis* sp. had the longest overall sting length (14.33 mm) though the relative sting length was 1, indicating that the overall length was likely related to the large size of the wasp. Velvet ants had a much larger relative sting length compared to most other wasps, with an outlying wasp, *Sapyga elegans*, grouping with the velvet ants (Fig. 1B).

We found a significant positive relationship between mesosomal length and sting length ($F_{2,57} = 111.2$; $R^2 = 0.796$, $P < 0.0001$) for both velvet ants and other aculeates (Fig. 1B). Additionally we found that there was a significant difference between velvet ants and other aculeates ($F_{2,57} = 111.2$; $R^2 = 0.796$, $P < 0.0001$). While both velvet ants and other aculeates show a positive relationship between mesosoma length and sting length, the significant interaction ($F_{3,56} = 107.7$; $P < 0.0001$) between velvet ant and non-velvet ant datasets indicates that velvet ants sting length increases significantly more as the velvet ants body size increases compared to all other aculeates (Fig. 1B).

While we did find a significant positive relationship between overall sting length and pain ($F_{1,58} = 18.75$; $R^2 = 0.2443$, $P < 0.001$), indicating that larger wasps generally have a more painful sting, we found no relationship between relative sting length (sting length compared to the mesosomal length) and pain ($F_{1,58} = 0.2682$; $R^2 = 0.0046$, $P = 0.6065$). Furthermore, we found marginally significant evidence that toxicity was inversely related to sting length ($F_{1,7} = 5.175$; $R^2 = 0.425$, $P = 0.05707$), indicating that aculeates with longer stings relative to their body size were less toxic than those with smaller stings relative to their body size (Fig. 2). Also, as has previously been suggested (*Schmidt, Blum & Overal, 1980*, *1986*), we found no significant relationship between venom lethality and reported pain of the sting ($F_{1,7} = 1.212$; $R^2 = 0.1476$, $P = 0.3073$). We did, however, find a weakly significant relationship between sociality and relative sting length ($F_{1,58} = 4.383$; $R^2 = 0.07026$, $P = 0.04068$), with social species having shorter stings relative to their bodies than solitary species. We found no relationship, though, between sociality and pain ($F_{1,58} = 2.848$; $R^2 = 0.04681$, $P = 0.09685$).

We found that the species' ecology, the host/prey use in particular, was significantly correlated to relative sting length ($F_{10,49} = 19.49$; $R^2 = 0.7991$, $P < 0.0001$) with those taxa that use immature Hymenoptera as their host having significantly longer stings compared to their bodies than all other aculeates (Fig. 3). These include all of the velvet ants and the sapygid wasp (a close relative of velvet ants (*Branstetter et al., 2017*)).

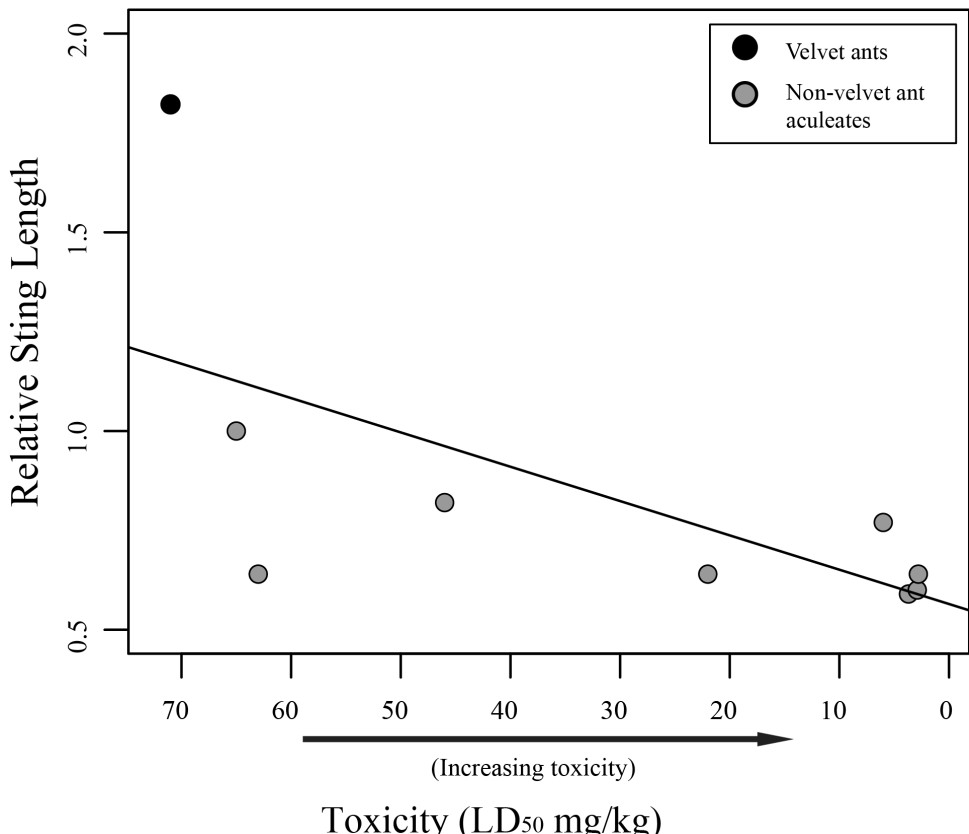

**Figure 2 Graph of relative sting length vs. toxicity.** Relative sting length is negatively correlated to toxicity. Relative sting length is a ratio of the sting length/mesosomal length and toxicity is measured in milligram per kilogram with lower numbers being more toxic. Velvet ants are marked with black and other aculeates are marked with gray. 

## DISCUSSION

Our results show that there is no link between relative sting length and pain. It should be mentioned that measuring the pain associated with stings is a subjective endeavor and these measures should be viewed as soft assessments rather than hard metrics. Regardless, while one might assume the longer stings would inflict more pain, based on personal observation, when someone is stung by an aculeate wasp, the sting only shallowly penetrates the skin. This observation suggests that the sting length is not used to inject venom deeper into the victim, but likely has been selected for other purposes (discussed below). While sting length does not seem to be associated with pain, we did find an inverse relationship between relative sting length and toxicity (though only a limited number of taxa could be included in the analysis), with taxa having shorter relative stings being more toxic. This could be related to the way different taxa use their stings. Most wasps use the sting to immobilize or kill their host (*Schmidt, 2016*). There is some necessity, therefore, for these wasps to evolve venoms that are toxic enough to effectively immobilize their prey. Velvet ants, on the other hand, parasitize hosts (immature hymenoptera) that are already immobile, and it has been suggested that their sting and associated venom is primarily used for defense (*Schmidt, 2016*). Highly toxic venom for defense might be

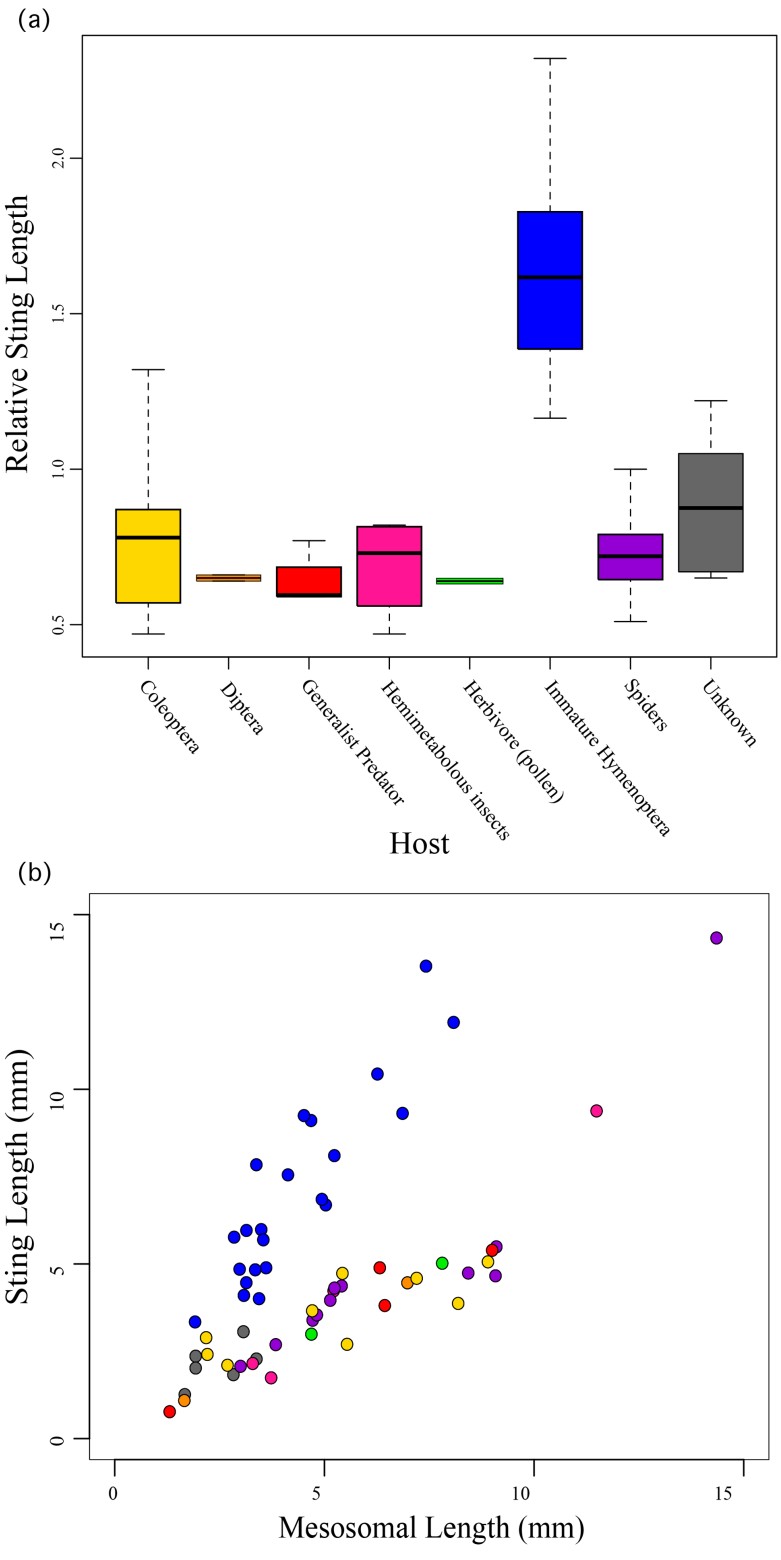

**Figure 3 Relative sting length compared to host preferences.** Relative sting length is significantly correlated to host use in those wasps that parasitize immature Hymenoptera, but not in all other aculeates. (A) Boxplot showing relative sting length vs. host use. (B) Scatter plot of sting length vs. mesosomal length (see Fig. 1) with taxa colored based on host preferences.

selected against, as it would be more beneficial for a velvet ant to inflict pain, but not mortally wound a potential predator facilitating learned avoidance, which has been demonstrated in feeding trials with various vertebrates (*Gall et al., in press*).

Our results clearly show that velvet ants have the longest stings out of the stinging wasps in relation to their body size (Figs. 1A and 1B). For example, the velvet ant *D. occidentalis* had a sting nearly as long as the tarantula hawk (*Pepsis* sp.) that was twice its size (Table 1). Some velvet ant species have been given the common name of "cow killer" (*Schmidt, 1990b*), which are theoretically named because anyone who was stung would claim it hurt bad enough to "kill a cow" (*Schmidt, 2016*). This ominous common name, however, is somewhat enigmatic given that velvet ants have some of the least toxic venoms of any of the wasps (*Schmidt, 1986b*; *Schmidt, Blum & Overal, 1980*, *1986*). While velvet ants have the longest sting (compared to their body) out of any aculeate, their stings are, however, short compared to many parasitic wasps. One of note being *Euurobracon yokohamae*, which has an ovipositor 7.7 times the length of the body (*Townes, 1975*).

It is not entirely clear why velvet ants have such long stings, yet the fact that sting length is correlated to host type suggests something about their parasitic nature has driven the evolution of exceptionally long stings. Below, we will explore two potential hypotheses that might explain the size of the velvet ant sting.

First, the long sting of velvet ants might help them immobilize their host in the tight confines of the host nest cell. While little is known about the behavior of velvet ants when they are in their host nest, it is clear that these wasps are parasitic primarily on immature bees and sphecid wasps, generally parasitizing the prepupa or pupal stage of their host (*Brothers & Finnamore, 1993*). Most velvet ants parasitize solitary, ground nesting species (*Brothers & Finnamore, 1993*). To successfully parasitize a host, the parasite must find the host nest, open the host nest cell (either underground or in a pre-existing cavity), and oviposit near the host prepupa or pupa. From the few descriptions of velvet ant parasitism behaviors it appears that the adult velvet ant opens the nest cell only enough to allow her head access, permitting her to probe the nest cell with her antennae to determine if the host has finished consuming its provision (indicating the nest cell is appropriate for oviposition) (*Brothers, 1972*). Once an appropriate nest cell is identified, the female velvet ant will turn around and insert the tip of the metasoma into the opening in the wall of the nest cell and probe around with her sting (*Brothers, 1972*). The female velvet ant will sting the host only if it is in the pupal stage, but will simply oviposit if the host is in the prepupal stage with the sting apparently serving to stop development of the host (*Brothers, 1972*; *Janvier, 1933*). It is possible that the long sting of the velvet ants enhances their ability to parasitize the host in the close confines of an underground nest cell.

This hypothesis, however, does not seem to be supported in other wasps. Scoliid wasps, for example, also parasitize ground nesting hosts, specifically beetle larvae (*Brothers & Finnamore, 1993*). While they undoubtedly also face similar challenges to the velvet ants in finding a host and paralyzing it with a sting in the tight underground burrow of the beetle, yet scoliids do not have a long sting. This suggests that other factors, other than

the tight confines of the host underground nest might be playing a role in the selective advantage of the length of the velvet ant's sting.

A second hypothesis regarding the length of velvet ant sting is that it evolved in response to predation pressures. Velvet ants are among the most highly defended of all stinging wasps (*Manley, 2000*; *Schmidt, 2016*; *Schmidt & Blum, 1977*). These defenses include aposematic coloration, stridulation (auditory aposematism), pungent exudate secretions, a hard cuticle, and a painful sting (*Manley, 2000*; *Schmidt, 2016*; *Wilson et al., 2012*). Not only do velvet ants have the longest stings (as our results clearly show), they also have one of the most flexible and maneuverable apical metasomal segments enabling them to reach their sting to nearly every part of their body (*Schmidt, 2016*). Because velvet ant's hosts are largely immobile, they are thought to only rarely sting their prey (*Schmidt, 2016*), instead it has been suggested their sting is primarily used to defend against predators (*Schmidt & Blum, 1977*). In fact, the length of the sting, combined with the hard cuticle of the velvet ants makes them nearly indestructible (*Vitt & Cooper, 1988*).

Several of the more unique aspects of the velvet ant sting and venom make them highly effective against predators. First, as is mentioned above, the length and agility of the sting, combined with their extraordinarily hard cuticle, aposematic coloration, and stridulation enables velvet ants to quickly and effectively train predators to avoid them. *Gall et al. (in press)*, for example, found that when a lizard (in this study the lizard was *Aspidoscelis tigris*) attacks a velvet ant, it is unable to crush it because of the hard cuticle, as the lizard attempts to manipulate the velvet ant in its mouth, the velvet ant is quickly able to sting the lizard. Once released, the aposematic coloration of the velvet ant apparently facilitates rapid learning in the lizard. In many instances after experiencing the sting of a velvet ant (and the other defenses) a lizard will not attempt to attack another velvet ant, even with over a year between exposures to these wasps (*Gall et al., in press*). The second aspect of velvet ant stings and venom chemistry that make them highly effective predatory deterrents is the mildly toxic, but highly painful sting. This enables velvet ants to train predators with a painful sting, but the mild toxin does no lasting damage.

The highly effective defenses of velvet ants, of which the long sting plays a key role, have enabled velvet ants to diversify around the world. Furthermore, these defenses have been influential in the evolution of the world's largest known Müllerian mimicry complex among diurnal velvet ants (*Wilson et al., 2012*, *2015*, *2018*).

## CONCLUSION

Our study of the sting length of various bees, aculeate wasps, and ants finds that velvet ants have the longest sting compared to their body size out of any aculeate. While there was no link between relative sting length and the pain associated with the sting, we did find an inverse relationship between relative sting length and toxicity, with taxa that have shorter relative stings being more toxic. While we found a significant relationship between host use and relative sting length, we suggest that the long sting length of the velvet ants may also be related to their suite of defenses to avoid predation.

## ACKNOWLEDGEMENTS

We thank Alex Kelley, Katie Weglarz, Nicole Boehme, Sarah Clark, Jake Jones, Andrew Ermer, Erik Pilgrim, David Tanner, Barry Webster, Edmund Williams, and Kevin Williams for help in collecting specimens.

### Funding

This research was supported by the Utah Agricultural Experiment Station, Utah State University, and approved as journal paper number 9092. The funders had no role in study design, data collection and analysis, decision to publish, or preparation of the manuscript.

### Grant Disclosures

The following grant information was disclosed by the authors:
Utah Agricultural Experiment Station, Utah State University.

### Competing Interests

The authors declare that they have no competing interests.

### Author Contributions

- Emily A. Sadler conceived and designed the experiments, performed the experiments, analyzed the data, prepared figures and/or tables, authored or reviewed drafts of the paper, approved the final draft.
- James P. Pitts conceived and designed the experiments, performed the experiments, analyzed the data, contributed reagents/materials/analysis tools, authored or reviewed drafts of the paper, approved the final draft.
- Joseph S. Wilson conceived and designed the experiments, analyzed the data, prepared figures and/or tables, authored or reviewed drafts of the paper, approved the final draft.

### Data Availability

The raw data and R code are provided as Supplemental Files.

### Supplemental Information

Supplemental information for this article can be found online at http://dx.doi.org/10.7717/peerj.4743#supplemental-information.

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
