# Peer review of "Stinging wasps (Hymenoptera: Aculeata), which species have the longest sting?"

_PeerJ, doi:10.7717/peerj.4743_

## Round 0.1 · original submission · Minor Revisions

Please address the comments made by the two reviewers either in the text or through a response. Thank you.

·

Basic reporting

The basic reporting is sound. A few more details are presented in the comments to authors panel.

Experimental design

The experimental design seems sound. A few minor comments are also listed in the comments to authors section below.

Validity of the findings

I find the validity of findings sound with the minor suggestion that the discussion on host biology is a bit weak and is really unnecessary. Its omission would not hurt the manuscript overall, rather might improve it.

Additional comments

The manuscript addresses some interesting questions relating to the importance of the aculeate hymenopteran sting length and various venom and defensive aspects of the insects. The presentation is mostly straightforward and readily understandable.

In the Methods section, the description of how the sting length is measured could be improved. The authors did not explain in detail how the length was determined; that is, was the length taken as the shortest distance between the triangular plate and the tip of the sting, or was some adjustment for the curvature and the length of the curve part of the sting measured? Many of the stings are curved, sometimes strongly curved as in the Mutillidae, and this length seems more suitable to use rather than simply the shortest route from one point to the next. When the sting is used it follows the shape of the sting shaft not the shape of the shortest distance between the triangular plate and the tip.

In the Results Table 1 only two ants are represented, yet ants are a crucially important group that are often highly toxic and painful and the study would benefit greatly by addition of a few more species – for example a Pogonomyrmex, an Odontomachus, and a Neoponera. These are all common species that should be readily available and would add important information to the study. This might also add more data to strengthen the evaluation of sting length versus venom painfulness and insect sociality.

Also for Table 1 the order would make more sense to have the taxa listed according to phylogeneny as presented in Branstetter et al 2017. This would help the reader visualize the importance of phylogeny versus biology and lifestyle.

Figure 1b would be improved with two changes: first, the axes need units; and second, since the Pompilidae are represented by numerous species and are an important comparison to the Mutillidae, I recommend making another line of a different color for them just as was done for the Mutillidae.

Figure 2 could be strengthened if how the points were obtained was clarified; that is, is each point the average for an entire family, or for an individual species? This might be important as some families have many representative taxa and others have only one.

I suggest the presentation relating to hosts and relative sting length be eliminated. This is because the many points from the Mutillidae, all of whom parasitize Hymenoptera, bias the overall relationship. Ideally we would compare equal numbers of taxa in each host or dietary type. I do not think eliminating this information would overall weaken the manuscript, rather would strengthen it.

Some of the references are missing journal volume and page numbers. These presumably will be added in the final version.

Reviewer 2 ·

Basic reporting

This manuscript presents data on sting length in the aculeate Hymenoptera, and interspecifically compares sting length to other aspects of wasp behavior and ecology to better understand sting length evolution. I think this is an interesting topic on an an interesting group of organisms. The paper is easy to read, and the figures are also easy to read and visually appealing. The literature is well-discussed. I have two major comments in this category:

1. The discussion is devoted to presenting interesting adaptive hypotheses explaining the long stings of velvet ants. However, it’s typical that the main results presented in the results section and figures are at least mentioned in the discussion. It seems that the discussion addresses the title question: Which species have the longest sting?, but overlooks the other questions asked with your comparative analyses. I would add a paragraph or two to discuss the relationship between (relative) sting length and toxicity, and the lack of a relationship between sting length and pain. When doing so, it would be important to point out that the pain estimates are likely rather noisy, given the method used. These patterns are mentioned in the abstract and the results section, and thus should be brought up in the discussion as well.

2. I was able to open and see the data and code in the supplemental files. However, I think it would be helpful to include an additional file explaining the columns in each file. For example, I don’t know what the column “kind” with values 1000 or 10 refers to. And why does each entry have a “velvet sting” and a “wasp sting” category? Unless I’m misunderstanding the dataset, each line in the sting_data.txt is a measured sample. If so, please include the species name, or further clarify what these data are.

Experimental design

I think the experimental design is OK, but has two drawbacks.

1. L134: The method of inferring pain scores seems prone to bias and noise. Were you going strictly by the nearest phylogenetic relative with an independent pain report? How many of the samples were “imputed” in this way? Given that species can vary in pain rather a lot even within genera (e.g. Pogonomyrmex), I think inferring pain levels from different FAMILIES is likely to be rather noisy. Pointing out that these data are likely to be noisy, and that this could influence the negative result you found, seems important.

2. Statistical power in phylogenetic comparisons
The authors report an association between sting length and host, as well as defensive behavior (last line of the abstract, and throughout the discussion). I think the adaptive hypotheses presented in the discussion (host use and defense) are good candidates to explain the longer stings of the velvet ants. However, it is also important to note that the sting length association with host type (and perhaps with defensive behaviors) is the result of a single phylogenetic origin in the velvet ants. Thus, the statistical evidence for an association with host type is less strong than assumed here. This doesn’t mean that the suggested hypotheses are wrong, but this limitation should be pointed out. In principle, any shared, derived feature of that group is as statistically associated with long sting length as host type. I don’t think it is necessary for this manuscript for the authors to perform a phylogenetically controlled statistical analysis, but it is important to acknowledge this weakness of the current analysis. I refer the authors to the immense literature on phylogenetic comparative analyses, which they likely are already familiar with (eg Harvey and Pagel 1991 The Comparative Method in Evolutionary Biology, Oxford University Press).

Validity of the findings

I think the findings are valid, especially if the comparative analyses are discussed in the discussion (see above), and a caveat is added to acknowledge limitations due to 1. inferred pain values and 2. a lack of phylogenetic control.

Additional comments

Minor comments
L19: inert a “While…” or an “…and…” or something else to make this read better.

L115: maybe clarify this a bit. The current wording made me think that you only have stings available, yet you somehow obtain mesosomal lengths…thus, sting slides are not the only available individuals. My guess is you did this to avoid destroying samples to get the sting measurements, but maybe make this clearer.


Table 1: i think it would be better if you say in the caption that pain data were not available for all taxa as well, and perhaps have another column indicating whether the pain score came directly from the literature or from personal experience, or inferred from related species.

L229: “ants” should be possessive, either ant’s or ants’, or singular (ant).

---

## Round 0.2 · accepted · Accept

Thank you for thoughtfully incorporating the reviewer comments.

#